# Multiplexed detection of febrile infections using CARMEN

M. Kamariza[1,2] ✉, K. McMahon[1], L. Kim[1], N. L. Welch[1,3], L. Stenson[1], L. T. Allan-Blitz[1,4,5], G. Sanders[4], P. Eromon[6], A. Muhammad[6], A. E. Sijuwola[6], O. O. Ope-ewe[6], A. O. Ayinla[6], C. l'Anson[1], I. Baudi[1], M. F. Paye[1], C. Wilkason[1], J. Lemieux[1,4], A. Ozonoff[1,3], E. Stachler[1], C. T. Happi[6,7] & P. C. Sabeti[1,8] ✉

Detection and diagnosis of bloodborne pathogens are critical for patients and for preventing outbreaks, yet challenging due to these diseases' nonspecific initial symptoms. We previously advanced CRISPR-based Combinatorial Arrayed Reactions for Multiplexed Evaluation of Nucleic acids (CARMEN) technology for simultaneous detection of pathogens on numerous samples. Here, we develop three CARMEN panels that target viral hemorrhagic fevers, mosquito-borne viruses, and sexually transmitted infections, collectively identifying 23 pathogens. We use deep learning to design CARMEN assays with enhanced sensitivity and specificity, validating them and evaluating their performance on synthetic targets, spiked healthy normal serum samples, and patient samples for *Neisseria gonorrhoeae* in the United States and for Lassa and mpox virus in Nigeria. Our results show multiplexed CARMEN assays match or outperform individual assay RT-qPCR in sensitivity, with matched specificity. These findings underscore CARMEN's potential as a highly effective tool for accurate pathogen detection for clinical diagnosis and public health surveillance.

Bloodborne pathogens (BBPs) cause major disruptions to human life and public health. While the prevalence of bloodborne viruses varies widely across the world[1], BBPs harbor some of the deadliest diseases driven by hemorrhagic fevers (e.g., Ebola, Lassa). These diseases are often difficult to diagnose because early symptoms, including fever, vomiting, and aches, are indistinguishable from each other[2–7]. Similarly, sexually-transmitted infections (STIs) (e.g., HIV, mpox, Hepatitis B) are endemic throughout the world, and present a particularly significant burden on morbidity and mortality in low- or middle-income countries (LMICs)[8–10]. Furthermore, such settings often lack the required tools to support pathogen diagnosis, and instead rely on syndromic management of symptomatic cases. Thus, accurate diagnostic testing is needed to control the disease spread.

The gold standard for diagnosing BBPs and STIs are polymerase chain reaction (PCR)-based methods, such as reverse transcription quantitative polymerase chain reaction (RT-qPCR), due to their high sensitivity and specificity[11]. However, these methods present several limitations, such as low throughput, high cost, and complexity[12–15]. In particular, the time-consuming nature of RT-qPCRs for each test has led to a growing interest in developing alternative diagnostic tools that are equally robust yet more cost-effective, and with increased throughput and multiplexing capabilities[16,17].

Moreover, an ideal diagnostic and surveillance technology should be capable of processing hundreds of patient samples simultaneously and detecting multiple pathogens concurrently. Indeed, circulating and emerging diseases require widespread deployability of molecular

[1]Broad Institute of MIT and Harvard, Cambridge, MA, USA. [2]Harvard Society of Fellows, Harvard University, Boston, MA, USA. [3]Department of Pediatrics, Harvard Medical School, Boston, MA, USA. [4]Division of Infectious Diseases, Department of Medicine, Massachusetts General Hospital, Boston, MA, USA. [5]Division of Global Health Equity: Department of Medicine, Brigham and Women's Hospital, Boston, MA, USA. [6]Institute of Genomics and Global Health (IGH), Redeemer's University, Ede, Osun State, Nigeria. [7]Department of Biological Sciences, College of Natural Sciences, Redeemer's University, Ede, Osun State, Nigeria. [8]Howard Hughes Medical Institute, Chevy Chase, MD, USA. ✉e-mail: mkamariz@broadinstitute.org; pardis@broadinstitute.org

tests for diagnosing and surveilling various pathogens. The recent SARS-CoV-2 pandemic emphasized the challenges of detecting known and potentially unknown coronaviruses, rapidly-evolving viral variants, as well as distinguishing related viruses that trigger similar symptoms[18,19].

CRISPR-based technologies, specifically the CRISPR-Cas13 system, have shown promise in detecting a diverse set of targets such as bacteria[20,21], leukemia[22], influenza subtypes[23], and viral infections, including COVID-19[24–26]. Platforms such as SHERLOCK and SHINE have been validated for viral diagnostics including SARS-CoV-2, Zika, and dengue viruses, while DETECTR has been applied to detect DNA viruses such as HPV, and SARS-CoV-2[27–30]. Both SHERLOCK and DETECTR have since advanced toward or received emergency use authorization (EUA) for clinical use, underscoring the growing maturity of CRISPR-based diagnostics in the field. The CRISPR-Cas13 assay is an RNA-targeting CRISPR enzyme system that can be programmed to detect specific RNA sequences, including viral RNA[25]. Upon recognition of the target RNA, the Cas13 endonuclease cleaves non-target RNAs, leading to a detectable fluorescent signal[24]. The CRISPR-Cas13-based Specific High Sensitivity Enzymatic Reporter UnLOCKing (SHERLOCK) diagnostic assay was shown to detect viral particles as low as 1.25 copies/μL, which in this instance was found to be of higher sensitivity compared to standard RT-qPCR[26,31]. Using SHERLOCK, researchers achieved 100% specificity and 100% sensitivity with a limit of detection (LOD) of 42 RNA copies per reaction using the SHERLOCK system on 154 clinical samples of SARS-CoV-2 RNA[28]. This system demonstrated robust performance by integrating an internal control for ribonuclease contamination which enhanced the test's suitability for resource-limited settings with an elevated risk of RNAse contamination. These Cas13-based approaches inherently offer multiplexing advantages, enabling targeting of a single viral genome at multiple sites, differentiation between related viruses or serotypes, and detection of different mutations within viral genomes[26,27,32]. Ultimately, the development of SHERLOCK established a foundation for accurate and scalable pathogen detection.

We previously combined the power of CRISPR-Cas13 detection of amplified targets with the multiplexing capacity of microfluidics to create Combinatorial Arrayed Reactions for Multiplexed Evaluation of Nucleic acids (CARMEN)[33]. Leveraging our ability to test thousands of assay-target pairs with CARMEN, we created an automated design algorithm using deep learning, Activity-informed Design with All-inclusive Patrolling of Targets (ADAPT). ADAPT can identify optimal guide sequences for detecting pathogens with high specificity and sensitivity[34]. We further developed a microfluidic CARMEN (mCARMEN) system that uses Standard BioTools' BiomarkHD instrument to enable automation and reduce labor[35]. This CRISPR-Cas13 detection system has enabled the multiplexed detection of SARS-CoV-2, influenza A and B viruses, along with six additional respiratory viruses plus one internal control assay in a single reaction[33]. It does so with greater sensitivity and 100% specificity compared to RT-qPCR, and reduces the risk of false-positive results through demonstrating minimal cross-reactivity to non-target RNAs[35]. Additionally, the mCARMEN assay can distinguish between closely related SARS-CoV-2 variants, such as Delta and Omicron, with high accuracy, making it a valuable tool for surveillance and epidemiological studies[35]. With its superior multiplexing capacity (simultaneously testing up to 192 samples against 24 targets), the (now collectively called) CARMEN assay allows for high-throughput analysis of clinical samples with reduced processing time and cost per test compared to traditional RT-qPCR methods.

In this paper, we set out to optimize the CARMEN workflow by further reducing processing time and using simplified instrumentation suitable for use in routine clinical laboratories. Using this optimized CARMEN platform, we develop and evaluate three BBP panels for large-scale and high-throughput detection and surveillance of viral hemorrhagic fevers (e.g., Ebola virus, Lassa virus), mosquito-borne

viral diseases (e.g., West Nile virus, Yellow fever virus,), and sexually transmitted illnesses (e.g., Mpox virus, Hepatitis B virus).

Using ADAPT[34], we design multiple crRNAs for each target to identify optimal guide sequences for detecting them with high specificity and sensitivity. Then, we test the condensed panels using synthetic materials and validate them with seedstocks, totaling 23 targets – 8 targets for BBP panel 1, 9 targets for panel 2, and 6 targets for panel 3 (each also including the human internal control, RNAse P). We assess the sensitivity and LOD of BBP assays using healthy normal serum (HNS) samples spiked with known concentrations of Lassa virus (LASV) and mpox virus (MPOX, formerly monkeypox virus) genomic materials. For the remaining targets, we conduct a proof-of-concept to show that the assays maintain sensitivity and specificity in clinical samples and serum background. Finally, we conduct validation tests using 10 *Neisseria gonorrhoeae* (Gon), 90 lassa virus (LAVS), and 42 mpox virus (MPOX) confirmed-positive patient samples, benchmarked against commercial RT-qPCR kits. Ultimately, we present a versatile diagnostic pipeline capable of facilitating high-throughput pathogen identification and microbial surveillance, enabling early detection and prevention of infectious disease outbreaks.

## Results

### Optimization of CARMEN workflow with Fluidigm's BiomarkX
We first sought to improve the CARMEN workflow to enable scalability for routine high-throughput and multiplexed analysis of patient samples in standard clinical laboratories. Previously, we described a microfluidic CARMEN (mCARMEN) system that uses Standard Bio-Tools' BiomarkHD instrument to enable automation and reduce labor[35]. Briefly, samples are extracted and amplified, followed by detection using integrated fluidic circuits (IFCs, Supplementary Fig 1). Specifically, these microfluidic IFCs enable highly multiplexed detection by combining 192 samples with 24 detection assays or 96 samples with 96 detection assays, depending on the IFC. Subsequently, fluorescence is measured on the Fluidigm BiomarkHD using automated protocols, capturing images every 5 min for 1–3 h at 37 °C. Taken together, this general workflow requires an estimated 5 to 6 h and, on a full chip, would cost an estimated $0.76 per sample per assay (see Supplementary Data 1).

To optimize this pipeline, we assessed whether the newer, simplified, model of the Biomark series, namely the BiomarkX, would improve the simplicity and workflow time of the BiomarkHD (Fig. 1A). Indeed, the BiomarkX demonstrated several advantages important for our goals: (1) a smaller footprint compared to the BiomarkHD, (2) a user-friendly interface with a single button to launch data acquisition, (3) a system that can be used with all types of IFCs, and (4) a shorter runtime (total workflow requires ~4 h). Together, these advantages deem the BiomarkX adaptable for any given purpose and suggest its easy deployment in clinical laboratories. We tested our previously designed respiratory virus panel (RVP) using the BiomarkX and evaluated detection of the corresponding nine respiratory virus synthetic DNA targets[35]. Specifically, synthetic materials at dilutions ranging from $10^4$ to $10^0$ copies/μL were amplified by RT-PCR, followed by CARMEN fluorescence detection using BiomarkX (Fig. 1B). We observed robust and sensitive detection of all nine targets included in the assay, demonstrating that detection performance was unaffected by the streamlined workflow with the BiomarkX.

### Assay design of bloodborne pathogen panels (BBPs)
To create each BBP panel, we strategically selected clinically-relevant microbes and grouped them based on their likelihood to co-infect or the extent to which their disease manifestations resembled one another (Supplementary Table 1). Notably, diseases such as Ebola, Marburg, and Lassa fever fall under the category of viral hemorrhagic fevers, which often present similar symptoms like headaches, abdominal pain, fever, nausea, and frequent progression to

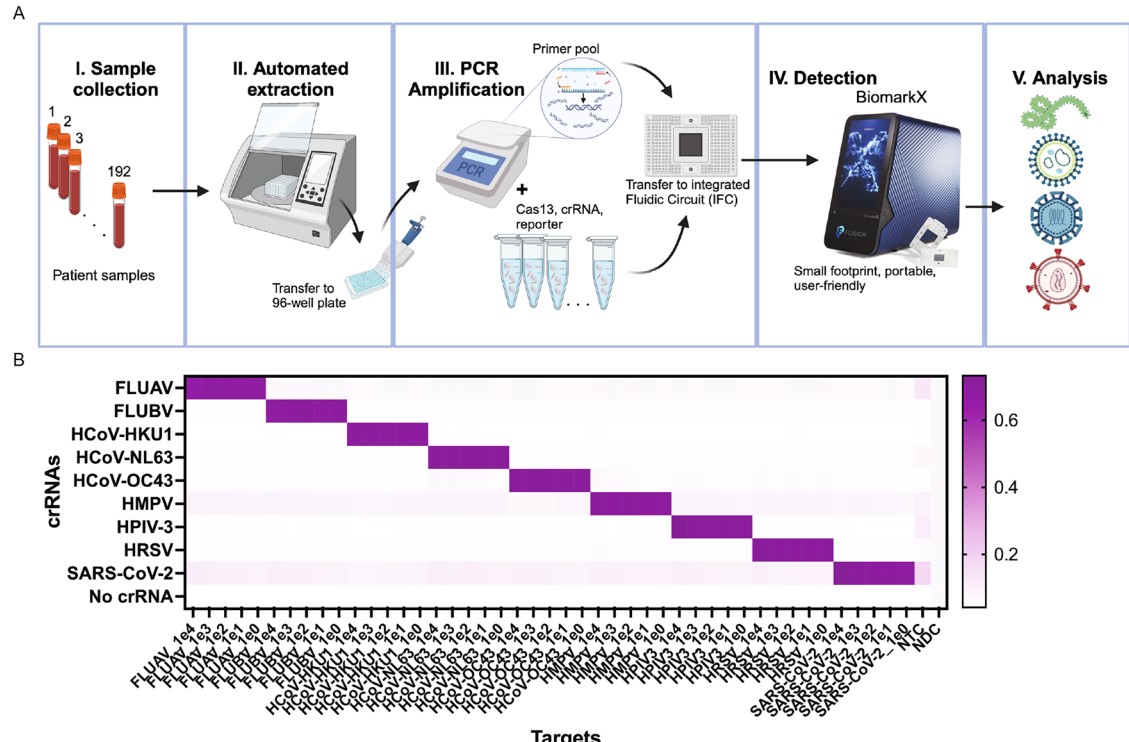

**Fig. 1 | Robust detection of the respiratory virus panel using the CARMEN-BiomarkX. A** Schematic of CARMEN workflow from blood samples to final results using the BiomarkX instrument from Standard BioTools. Created in BioRender. Kamariza, M. (https://BioRender.com/98chjgc). **B** Heatmaps illustrating CARMEN assay performance for the respiratory virus panel (RVP) including SARS-CoV-2, HCoV-HKU1, HCoV-OC43, HCoV-NL63, FLUAV, FLUBV, HPIV-3, HRSV, HMPV. Data shows fluorescence intensity of synthetic DNA fragments, at $10^4$-$10^0$ copies per µL, and corresponding vCiral Cas13 crRNAs at 1-hour post-reaction initiation. NTC No Target Control. NDC No Detection Control (no Magnesium).

hemorrhagic bleeds[36,37]. Accordingly, we included these symptomatically similar pathogens in a single panel, BBP1, alongside other fever-causing viruses (Supplementary Table 2). Similarly, we developed panels BBP2 and BBP3 to target mosquito-borne viral diseases and sexually transmitted infections (STIs), respectively.

Ultimately, we designed assays to detect 23 pathogens across three distinct panels, including 8 targets for viral hemorrhagic fevers (BBP1), 9 targets for mosquito-borne viral diseases (BBP2) and 6 targets for STIs (BBP3). BBP1 contained (1) Crimean congo hemorrhagic fever virus (CCHFV), (2) Zaire ebolavirus (EBOV), (3) Human immunodeficiency virus 1 (HIV1), (4) Human immunodeficiency virus 2 (HIV2), (5) Lassa virus (LASV), (6) Marburg virus (MBV), (7) *West nile virus* (WNV), (8) Yellow fever virus (YFV). BBP2 contained (9) Chikungunya virus (CHI), (10) Dengue virus (DENV), (11) Hantaan orthohantavirus (HTV), (12) *Measles morbillivirus* (MMV), (13) Rabies lyssavirus (RBV), (14) Rift Valley Fever virus (RVFV), (15) Zika virus (Zika), (16) Nipah henipavirus (NPV). BBP3 contained (17) mpox virus (MPOX, formerly monkeypox virus), (18) Hepatitis B virus (HBV), (19) Hepatitis C virus (HCV), (20) Herpes Simplex Virus Type 2 (HSV2), (21) *Chlamydia trachomatis* (Chla), (22) *Treponema pallidum subsp. pallidum* (Syph), and (23) *Trichomonas vaginalis* (Trich) (Supplementary Table 2). Each panel also includes RNAseP as an internal positive control.

For each target, we used ADAPT to design five microbe-specific crRNA assays and PCR primer pairs, totaling 115 assay designs[34]. This involved collecting complete genomes from NCBI and utilizing FASTA files to generate compatible crRNAs and primer sequences, as predicted by the ADAPT program. While most of ADAPT's designs included a single crRNA guide, LASV and DENV required multiple crRNA designs to maximize coverage across all analyzed genomes and ensure efficient capture of genomic diversity[27,38]. Indeed, this multi-guide system is predicted to capture LASV-Sierra Leone and LASV-Nigeria clades, as well as all four DENV serotypes (Supplementary Fig 2, 3). Our

sequence analysis demonstrated that ADAPT-designed assays represented between 85 and 100% of the complete genomes collected from NCBI for all the pathogens, suggesting their effectiveness in detecting the intended target sequences (Supplementary Table 3).

**BBP CARMEN development and testing using synthetic material**
Using synthetic DNA targets at a concentration of $10^8$ copies/µL, we conducted CARMEN experiments to identify which crRNA assay yielded the highest fluorescent signals over background ratio for each target (Supplementary Figs. 4–6). We prioritized assays that combined high specificity with strong sensitivity and selected the best-performing candidates for downstream analyses. After down-selection, we integrated these optimized assays into their respective panels and evaluated their collective performance for multiplexed detection of BBP1, BBP2, and BBP3 pathogens (Fig. 2A–C). In line with previous CARMEN studies, our results demonstrated no off-target signals, indicating compatibility and suitability for the multiplexed detection of BBPs.

We then sought to characterize the LOD values for each of the 23 targets in the BBP panels. We prepared dilutions of synthetic materials ranging from $10^5$ to $10^1$ copies/µL for amplification, followed by CARMEN fluorescence detection (Fig. 2D–F). We observed variable LODs across the 23 targets. Notably, for BBP1 and BBP3, all targets exhibited LODs of $10^3$-$10^1$ copies/µL (Fig. 2D, F). Many targets, such as EBOV, LASV, and Zika, demonstrated significant fluorescence even at the lowest dilution tested (10 copies/µL), indicating highly sensitive detection. In contrast, BBP2 exhibited higher LODs, which has been associated with lower assay performance[17]. This difference is likely due to the higher number of pooled assays in BBP2 compared to the other two panels. Nevertheless, with the exception of CHI, HCV, and RBV, the remaining targets in BBP2 still demonstrated LODs of $10^3$-$10^1$ copies/µL. These results demonstrate sensitive multiplexed detection of

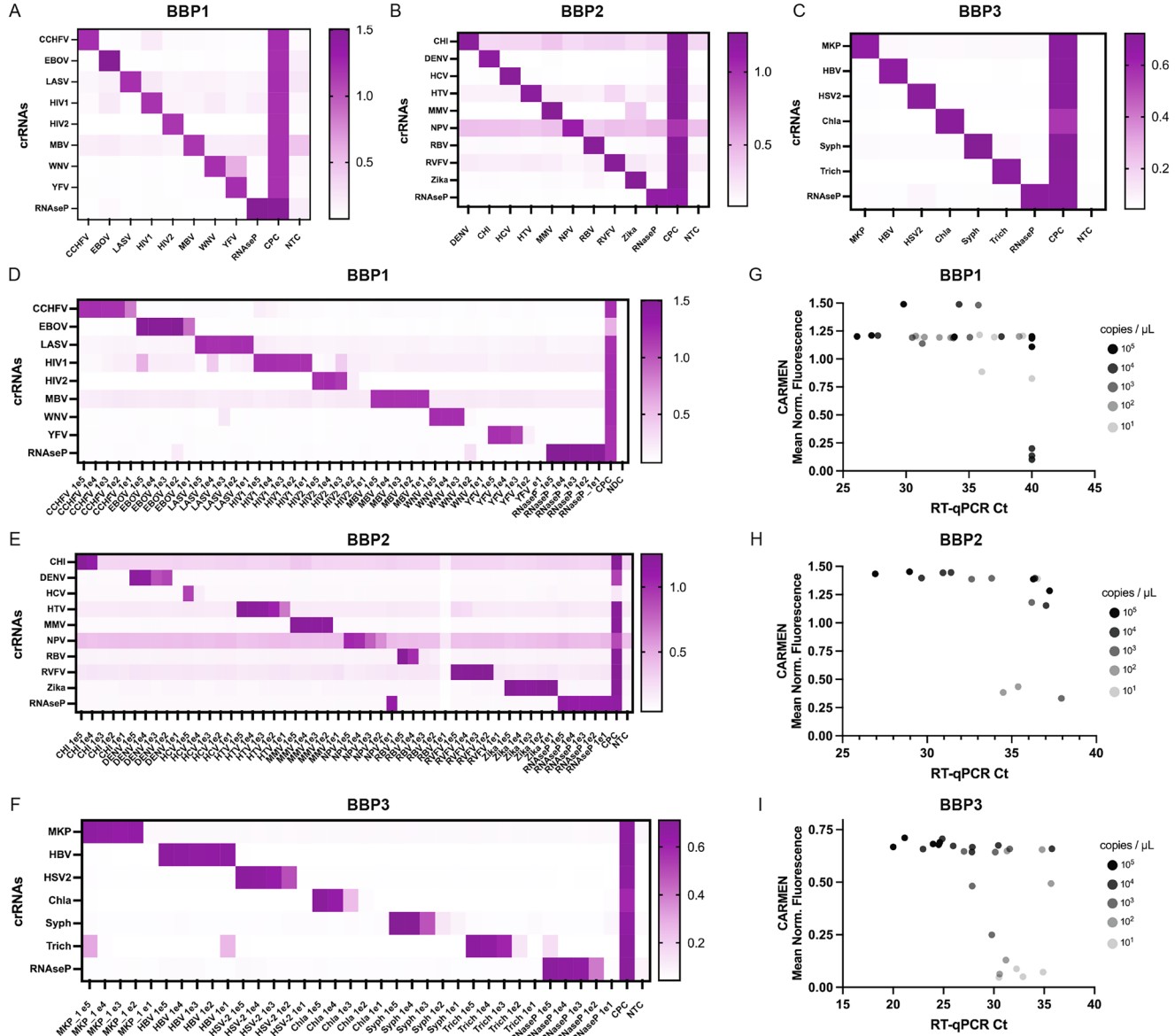

**Fig. 2 | Sensitivity and specificity assessment of CARMEN assays for Bloodborne Pathogen Panels (BBPs) synthetic detection. A–C** Heatmaps illustrating CARMEN assay performance for (**A**) BBP1, (**B**) BBP2, and (**C**) BBP3 panels, showing fluorescence intensity of synthetic DNA fragments, at $10^8$ copies/μL, and corresponding viral Cas13 crRNAs at 1-hour post-reaction initiation. **D–F** Heatmaps showing fluorescence values of the targets at the indicated concentrations (ranging from $10^5$ to $10^1$ copies/μL) for (**D**) BBP1, (**E**) BPP2, and (**F**) BBP3. Normalized fluorescence signal at 1 h post-reaction initiation of corresponding viral Cas13 crRNAs is shown. NTC No Target Control. CPC Combined Positive Control. NDC No Detection Control. **G–I** Comparison analysis of CARMEN fluorescence to RT-qPCR Ct values for (**G**) BBP1, (**H**) BBP2, and (**I**) BBP3.

synthetic BBP samples with minimal off-target signals, capable of detecting target sequences.

We compared the performance of CARMEN fluorescence LODs for individual assays within our multiplexed panels against an RT-qPCR comparison assay for singular targets. Using the same synthetic materials at dilutions ranging from $10^5$ to $10^1$ copies/μL, we observed a general trend of higher LODs (i.e., lower sensitivity) for qPCR when compared with corresponding CARMEN results (Fig. 2G–I, and Supplementary Fig 7). Specifically, we observed sensitive fluorescent detection at concentrations of 10 copies/μL for BBP1 despite yielding Ct values of 35 or higher (Fig. 2G, and Supplementary Fig 7A). While BBP2 exhibited lower performance for the NPV assay, resulting in higher CARMEN LODs compared to RT-qPCR LODs (Supplementary Fig 7B), the remaining assays demonstrated increased sensitivity using CARMEN compared to RT-qPCR analysis (Fig. 2H). For BBP3, we

observed slightly higher CARMEN LODs, particularly for Chla and Syph, which may be attributable to the relatively higher GC content of bacterial species (Fig. 2I, S7C). Nonetheless, on average, CARMEN detection outperformed RT-qPCR analysis (Supplementary Fig 7D).

Thus, comparison with RT-qPCR analysis consistently showed CARMEN's superior performance, highlighting its potential for multiplexed BBP detection.

### Specificity and sensitivity analysis of BBP detection using Healthy Normal Serum samples

We evaluated the specificity of BBP detection using clinical samples obtained from healthy individuals. We purchased twenty (20) serum samples labeled as Healthy Normal Serum (HNS) from Boca Biolistics, each containing one milliliter of serum. Following standard extraction and RT-PCR protocols, we assessed whether BBP CARMEN detection

resulted in background fluorescence using these samples (Supplementary Fig 8). After thresholding against the NTC, with the exception of Trich, we found no false positive results across all three BBP panels for all 20 HNS samples, consistent with RT-qPCR data (Supplementary Fig 8A–C). Thus, Trich was eliminated from BBP3 for downstream analyses (Supplementary Fig 8D). Next, we tested either commercially available patient samples, commercial gRNA/DNA spiked into HNS, or synthetic RNA/DNA of the various targets spiked into HNS at a final concentration of $10^5$ copies/uL per sample (see Supplementary Data 1 for detailed list). Following extraction and standard CARMEN, we observed robust and selective fluorescence signal with minimal off-target effects for all three panels (Fig. 3A–C, S8E-G), demonstrating high performance in complex clinically-relevant samples.

Subsequently, we investigated whether the LODs determined using synthetic samples were affected by the extraction step upstream of RT-qPCR and detection analysis (Fig. 3D–K, S9–S11). We utilized pooled HNS samples, subsequently aliquoting and spiking them with known concentrations of LASV or MPOX genomic DNA or RNA. We first performed initial optimization analysis with synthetic LASV and identified 50 nM concentration as the optimal RNAse P primer concentration (Supplementary Fig 9). Next, we spiked LASV and MPOX viral genomic material (RNA for LASV and DNA for MPOX) at final concentrations of $10^4$ to 1 copies/µL and $10^3$ to 1, respectively, into each HNS aliquot, followed by extraction of nucleic acid materials using the Zymo Research Quick-DNA/RNA MagBead™ extraction kit (R2130/R2131) on the KingFisher Flex instrument. We then used extracted material as input into the Aldatu Biosciences PANDAA LASV (2011096) or RayBiotech MPXV (PCR-MPXV) RT-qPCR detection kits, and standard CARMEN analysis.

Our results demonstrated specific and sensitive detection of BBP targets, with an improved sensitivity compared to RT-qPCR, and high concordance between the two assays. We found 100% of LASV and 95% of MPOX-contrived HNS samples positive using CARMEN, including samples at concentrations of -1 copy/µL (Fig. 3D, E). Under these conditions, the LOD was determined to be as low as -1 copy/µL, with notable effects of the extraction step on detection sensitivity. We observed a robust CARMEN signal, albeit with a single WNV contaminant, even at the lowest concentration tested (Supplementary Fig 10A). Using these same samples, we found 96% of LASV and 80% of MPOX contrived HNS samples positive using RT-qPCR (Fig. 3F, G, and Supplementary Fig 10B-C). By comparing these results with CARMEN, we found 100% and 96% concordance between the two assays for LASV and MPOX, respectively (Fig. 3H, I). Finally, we spiked LASV and MPOX genomic material at either $10^1$ or $10^0$ into 20 HNS aliquots followed by comparative CARMEN and RT-qPCR analysis. We found that, at the lowest $10^0$ copies/µL, all 20 LASV and MPOX samples were positive by CARMEN, but not by PANDAA LASV or RayBiotech MPXV kits, which identify only 10 and 17 positive samples, respectively (Fig. 3J, K, and Supplementary Fig 11).

These findings underscore the reliability of BBP CARMEN as a robust assay for the accurate and sensitive detection of microbial pathogens in human samples.

### Validation of BBP CARMEN assays using patient samples

We assessed whether our BBP-CARMEN platform could successfully detect pathogens from confirmed-positive patient samples in hospital and field settings (Fig. 4). We partnered with the Massachusetts General Hospital (MGH) Sexual Health Clinic and obtained 10 patient samples with confirmed-positive diagnoses for *N. gonorrhoeae* (Gon), as well as 38 Gon negative samples, by standard clinical diagnostic tests. Of these 38 negative samples, three were also annotated as *Chlamydia trachomatis* (Chla) positive (Chla is included in BBP3). We had previously shown that this CRISPR-Cas13 assay performed well on clinical samples in a single assay design[39,40]. We conducted CARMEN detection on these patient samples and analyzed the data using the above-mentioned threshold calculations.

All ten confirmed-positive samples were found positive for Gon using BBP3-CARMEN (Fig. 4A) yielding 100% positive concordance. Serendipitously, one confirmed-positive sample was also found positive for HSV2 (included in the same BBP3 panel). Of note, one confirmed-negative Gon sample was found positive by CARMEN, and the remaining 37 Gon negative samples were found negative for Gon, yielding a 97% negative concordance rate (Supplementary Fig 12). Interestingly, of the three Chla positive samples, only two were found positive for Chla using CARMEN (Fig. 4B). For Gon samples, our results demonstrated 100% concordance between BBP3-CARMEN and the diagnostic test results from the MGH Sexual Health Clinic (Fig. 4C).

Finally, to demonstrate versatility and applicability of our system in low-resource settings, we validated our BBP CARMEN assays in Nigeria using stored patient samples with confirmed-positive diagnoses of LASV and MPOX. In total, we extracted 90 LASV and 42 MPOX-positive samples previously identified at the hospital. Of note, the extended storage duration of the samples raised the potential for sample degradation. We thus amplified and analyzed the samples following standardized operating procedures and optimizations based on contrived sample testing using BBP CARMEN alongside simultaneous RT-qPCR testing ((Aldatu Biosciences PANDAA LASV RT-qPCR for LASV and RayBiotech Mpox virus (MPXV) PCR Nucleic Acid Detection Kit for MPOX)) (see Supplementary Information for specific SOP details).

The CARMEN assay demonstrated sensitivity that outperformed RT-qPCR for LASV detection and matched RT-qPCR for MPOX detection. CARMEN identified 31 LASV-positive samples (34%) and 39 MPOX-positive samples (93%) (Fig. 4D, E), compared to RT-qPCR, which identified 27 LASV-positive samples (30%) and 39 MPOX-positive samples (93%) (Fig. 4F, G). For the LASV samples, the overall low positivity rate for both CARMEN and RT-qPCR assays suggests that sample integrity may have been compromised during storage or retrieval. Despite this limitation, all 27 LASV-positive samples identified by RT-qPCR were also detected by CARMEN (Fig. 4H). An additional four LASV samples were detected by CARMEN, but not by RT-qPCR, demonstrating a 96% concordance rate and highlighting the enhanced sensitivity of the CARMEN assay in identifying cases that may be missed by standard PCR-based diagnostic methods. These data align with our results from synthetic and contrived samples, as well as prior studies on clinical samples[35]. Similarly, for MPOX detection, the two assays achieved 100% concordance, with 39 positive and 3 negative results (Fig. 4I).

## Discussion

The development of BBPs for CARMEN detection represents a crucial step toward addressing the diagnostic challenges posed by illnesses sharing similar clinical symptoms. By strategically selecting microbes with overlapping symptoms, we designed three panels broadly targeting: viral hemorrhagic fevers, mosquito-borne viral diseases, and sexually transmitted infections. The comprehensive design of 23 CRISPR-Cas13 assays using the ADAPT program ensured broad coverage of target sequences across diverse genomes, reflecting the effectiveness of our assay design strategy.

Subsequently, using synthetic materials, we characterized the LOD for each target in the BBP panels. Our results demonstrated sensitive detection of BBP targets, with many exhibiting LODs of 1,000 copies per microliter (copies/µL) or less. Notably, CARMEN fluorescence LODs outperformed RT-qPCR Ct analysis in terms of sensitivity, highlighting the potential of CARMEN for sensitive multiplexed BBP detection. However, variations in LODs across targets and panels underscore the importance of assay optimization and validation for each specific application. Furthermore, specificity analysis using HNS samples confirmed the absence of false-positive results across all BBP panels, validating the specificity of our assays. The compatibility of CARMEN with patient samples was further validated through analyzing contrived HNS samples spiked with known concentrations of viral

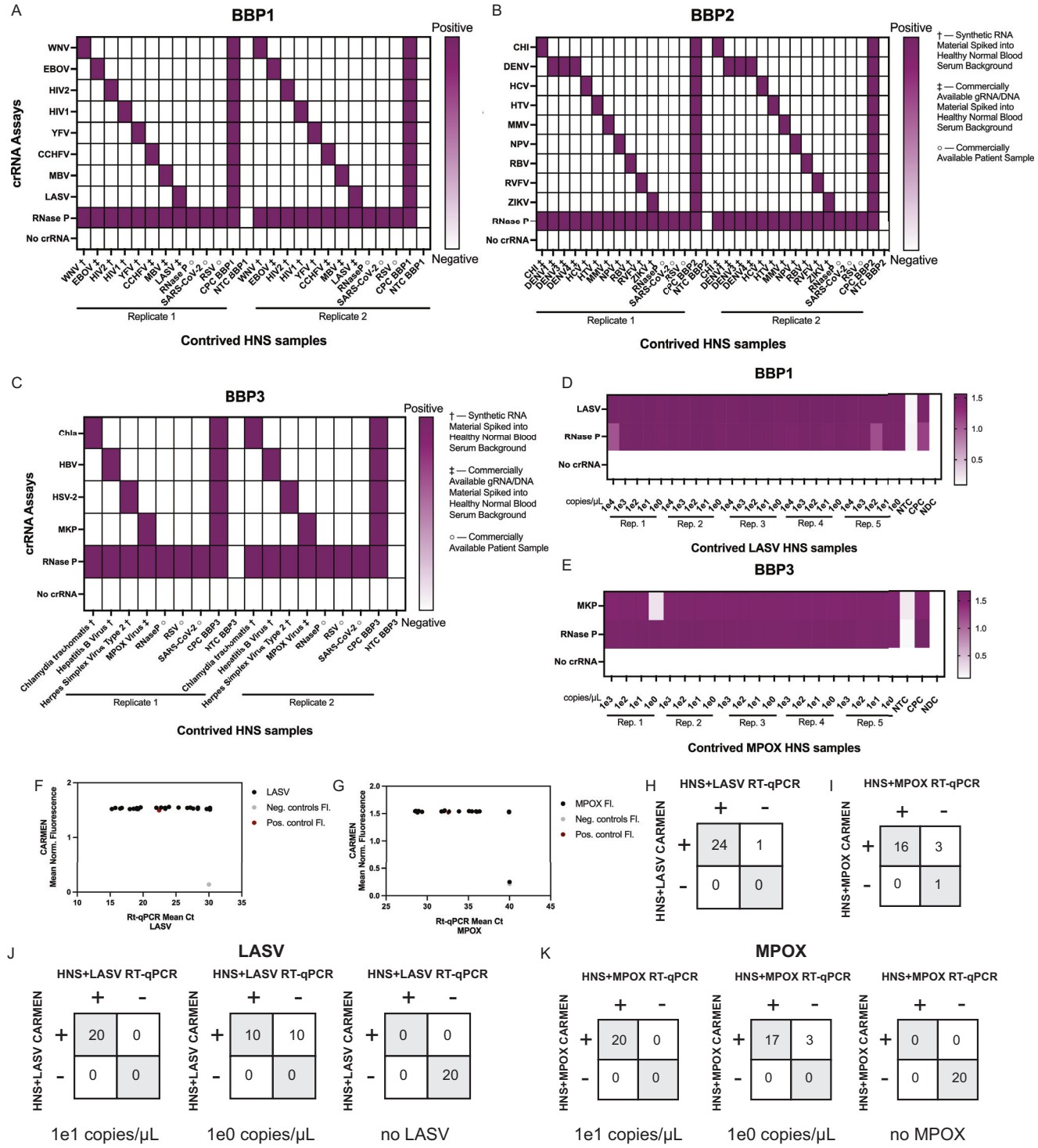

**Fig. 3 | Sensitivity and specificity analysis with contrived healthy normal serum (HNS) samples. A–C** Heatmaps showing positive-negative results of target synthetic RNA spiked into HNS at final concentration of $10^5$ copies/μL, commercial gRNA/DNA spiked into HNS, or commercial patient samples (as indicated) followed by CARMEN detection with (**A**) BBP1, (**B**) BBP2, or (**C**) BBP3 panels. Positive threshold for each assay was determined using the following calculation: (average(NTC Fl.) + 3*(standard deviation(NTC Fl.)). (**D, E**) Heatmaps showing fluorescence values of (**D**) LASV or (**E**) MPOX RNA extracts spiked into HNS at the indicated concentrations (ranging from $10^3$ to $10^0$ copies/μL) followed by CARMEN detection with BBP1 and BBP3 respectively. For all heatmaps, normalized fluorescence signal at 1 h post-reaction initiation of corresponding viral Cas13 crRNAs is shown. NTC

No Target Control. CPC Combined Positive Control. NDC No Detection Control. **F, G** Comparison analysis of CARMEN fluorescence to RT-qPCR Ct values for contrived (**F**) LASV and (**G**) MPOX HNS samples. **H, I** Concordance table of CARMEN fluorescence to RT-qPCR Ct values for contrived (**H**) LASV and (**I**) MPOX HNS samples at varying concentrations (ranging from $10^3$ to $10^0$ copies/μL). **H, I** Concordance table of CARMEN fluorescence to RT-qPCR Ct values for five replicates of contrived (**H**) LASV and (**I**) MPOX HNS samples at varying concentrations (ranging from $10^3$ to $10^0$ copies/μL). **J, K** Concordance table of CARMEN fluorescence to RT-qPCR Ct values for 20 replicates of contrived (**H**) LASV and (**I**) MPOX HNS samples at $10^1$ or $10^0$ copies/μL.

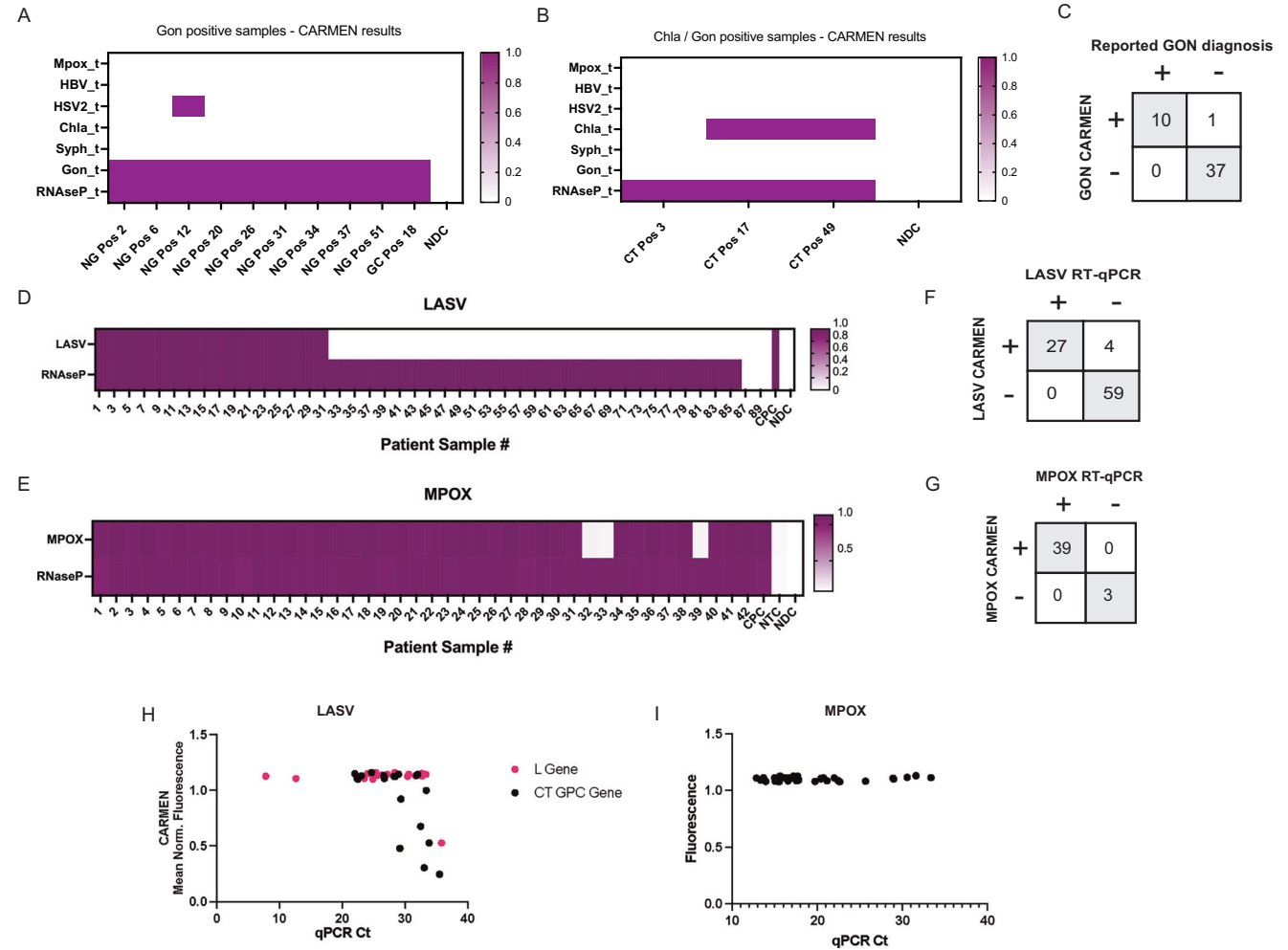

**Fig. 4 | Sensitive, accurate, detection of Gonorrhea, Chlamydia, Lassa virus, and Monkeypox in patient samples. A, B** Heatmaps illustrating CARMEN detection of (**A**) *Neisseria gonorrhoeae (Gon)*, and (**B**) *Chlamydia trachomatis (Chla)*. **C** Comparison analysis of CARMEN fluorescence to RT-qPCR Ct values for Gon patient samples. **D, E** Heatmaps showing fluorescence values of (**D**) LASV or (**E**) MPOX clinically-positive patient samples. For all heatmaps, normalized

fluorescence signal at 1 h post-reaction initiation of corresponding viral Cas13 crRNAs is shown. CPC Combined Positive Control. NDC No Detection Control. **F, G** Concordance table of CARMEN fluorescence to RT-qPCR Ct values for (**F**) LASV and (**G**) MPOX patient sample. **H, I** Comparison analysis of CARMEN fluorescence to RT-qPCR Ct values for (**H**) LASV, and (**I**) MPOX.

seedstocks, which demonstrated specific and sensitive detection of BBP targets even in complex sample matrices.

Lastly, validation using patient samples with confirmed diagnoses of Gon, LASV, and MPOX infections highlighted the robustness of the BBP CARMEN assays in real-world clinical settings. The discrepancies observed between the original tests and the current RT-qPCR results for LASV samples underscore the potential impact of long-term storage on sample integrity. Because these samples were obtained from a previously collected cohort without accompanying sequencing data, the discrepancy could not be conclusively validated. However, the high concordance observed between CARMEN and RT-qPCR and the greater positive detection using the CARMEN assay highlight the reliability of CARMEN, with increased sensitivity, for bloodborne pathogen detection. The high pathogenicity of BBP targets presents a substantial obstacle in procuring suitable seedstocks and positive patient samples essential for panel validation. Our future research will aim to address this challenge through the establishment of international partnerships and efforts to create sample repositories.

The optimization of the CARMEN workflow using Standard Biotool's BiomarkX also represents a significant advancement toward enabling scalable, high-throughput, and multiplexed analysis of patient samples in standard clinical laboratories. By transitioning from the

BiomarkHD to the BiomarkX instrument, we aimed to streamline the workflow and reduce the turnaround time of diagnostic testing. Indeed, the BiomarkX offers several advantages over its predecessor, including a smaller footprint, user-friendly operation, and compatibility with various IFCs. Our evaluation of the BiomarkX instrument for detecting synthetic DNA from respiratory viruses demonstrated robust performance, validating the suitability of the BiomarkX for CARMEN analysis. More broadly, the ease of transition of our CRISPR-based assays to these new platforms points to the robustness of these assays, and the opportunities to integrate them into a range of technological platforms.

Ultimately, this study demonstrates the successful optimization and validation of CARMEN assays for sensitive and specific detection of bloodborne pathogens, laying the foundation for their broader application in clinical diagnostics and public health surveillance. CARMEN offers a major advantage over other CRISPR-based assays by enabling the simultaneous detection of up to 24 targets—far more than the single to few pathogens typically targeted by earlier methods such as SHERLOCK, DETECTR, and SHINE. In addition, CARMEN can resolve viral diversity at the species, strain, and even single nucleotide polymorphism (SNP) level, to deliver high diagnostic precision. CARMEN is complementary with next-generation sequencing which can enhance its capacity for pathogen detection, discovery, and evolutionary

analysis, making it well-suited for large-scale and in-depth pathogen surveillance. However, the widespread implementation of CARMEN in low-resource settings will require addressing key practical challenges, notably the requirement for specialized equipment (BiomarkX), adequate infrastructure (e.g., cold chain for reagents), standard clinical laboratory resources to minimize the risk of contamination in high-throughput workflows, and the need for robust assay standardization and data analysis tools. Further, streamlined nucleic acids extraction protocols, along with the integration of isothermal amplification or amplification-free methods into the CARMEN workflow, would enable decentralized applications in settings where power and equipment access are constrained. In addition, while CARMEN demonstrated high specificity across all panels, including in cross-reactivity testing and healthy serum controls, Cas13a enzymes may tolerate certain mismatches, and future work will incorporate sequencing-based validation and computational off-target analysis to further characterize crRNA specificity and resolve any discordant clinical results. Lastly, future clinical studies should incorporate powered sample sizes, blinding, and more advanced statistical analyses to further support regulatory evaluation and deployment. Addressing these challenges alongside expanding the range of detectable pathogens will enable the widespread clinical utility of CARMEN in diverse settings. Ultimately, the adoption of CARMEN technology has the potential to transform pathogen detection and facilitate early intervention strategies for infectious disease control and management.

## Methods

### Statistics & reproducibility
No statistical method was used to predetermine sample size. No data were excluded from the analyses. The experiments lacked randomization, and the investigators were not blinded to allocation during both experiments and outcome assessment.

### MGB patient samples
Stored clinical urine specimens were previously collected between March and September 2023 as a part of an ongoing clinical trial in the Massachusetts General Brigham (MGB) Sexual Health Clinic (NCT05564299). Participants were selected using the following criteria: 18 years or older, presented with symptoms of urethritis or cervicitis (urethral or virginal discharge, dysuria, or dyspareunia), not known to be pregnant at the time of enrollment, no known exposure to *N. gonorrhoeae* or *Chlamydia trachomatis* within the previous six weeks, no concurrent symptoms at extragenital sites, and were willing to provide informed consent. 1.5 mL of each urine sample were aliquoted into cryotube vials, assigned a unique study identifier, and stored at -80 °C within 30 min of sample aliquoting. As a part of routine care, confirmatory diagnostic testing included gram stain, nucleic acid amplification testing, and culture with susceptibility testing using standard methods. These specimens were deemed to be *N. gonorrhoeae* positive if standard clinical diagnostics showed positive results.

### IGH Patient samples statement
Patient samples in Nigeria were selected based on PCR outcome: confirmed Positive, with Ct values ranging from 15-40. Plasma and swab samples, for LASV and MPOX respectively, were collected in EDTA tubes for further processing. Samples were stored at -80 °C and at -196 °C liquid nitrogen for up to one year in plasma AVL for LASV, and Viral Transport Media for MPOX. Prior to CARMEN analysis, samples were processed by inactivation with AVL and extraction via AviPure Automated Extraction Kit (Avicenna) on the AviPure AX96™ (Avicenna).

### General procedures.
A detailed standard operating procedure for running the BBP panels on CARMEN can be found in the Supplementary Information.

### Production and processing of synthetic materials.
Synthetic materials were synthesized and handled as described before[35]. Briefly, crRNAs (Integrated DNA Technologies) were resuspended in nuclease-free water to 100 μM and then diluted for input into the detection reaction. Primer sequences (Eton or Integrated DNA Technologies) were also resuspended to 100 μM in nuclease-free water and combined at specified concentrations for pooled amplification.

### In-vitro transcription (IVT) of DNA targets.
Synthetic DNA gene fragments of RNA viruses were in-vitro transcribed to use as RNA standards for experiments as described previously[35]. Briefly, DNA targets were procured from Integrated DNA Technologies and subjected to in-vitro transcription (IVT) using the HiScribe T7 High Yield RNA Synthesis Kit (New England Biolabs). Transcriptions adhered to the manufacturer's recommendations, with a 20 μL reaction volume (30 μL if the target of interest was above 300 base pairs in length), incubated overnight at 37 °C and followed by DNAse I (New England BioLabs) treatment to remove the DNA template. Purification of transcribed RNA products employed RNAClean XP beads (Beckman Colter), and quantification was carried out using the Invitrogen™ Qubit™ RNA High Sensitivity (HS) kit as recommended by the manufacturer. For experimentation purposes, RNA was serially diluted from $10^8$ down to $10^1$ copies/μL and utilized as input for the subsequent amplification reaction.

### Nucleic acid extraction and processing.
Samples underwent automated total nucleic acid extraction utilizing the Zymo Research Quick-DNA/RNA™ MagBead kit on the KingFisher Flex 96 Deep-well Head Magnetic Particle Processor (Thermo Fisher Scientific). Following the manufacturer's instructions, samples were pre-processed based on their sample matrix type as outlined in Sample Preparation in the Quick-DNA/RNA™ MagBead protocol provided by Zymo Research (pages 6–9, catalog nos. R2130/R2131). 200 μl of sample material produced from pre-processing was inputted into the extraction pipeline as outlined in Automation Reference Guide–KingFisher Flex in the Quick-DNA/RNA™ Magbead – Co-Purification protocol provided by Zymo Research (pages 3–4, catalog nos. R2130/R2131). Sample material was then placed on the KingFisher Flex instrument alongside required reagents and the KingFisher Flex protocol R2130_Quick DNARNA Magbead_KingFisherFlex_Copurification_v2.bdz was run (provided by Zymo Research). RNA and/or DNA from the input sample material was eluted in 50 μl of nuclease-free water and either added as input into the RT-PCR amplification step or stored at -80 °C until usage.

### QIAGEN OneStep RT-PCR amplification.
Reverse-transcription PCR amplification was performed as described previously[35]. Briefly, a total reaction volume of 50 μl was used: 12.5 μl QIAGEN OneStep RT–PCR 5x buffer, 3 μl each of pooled target forward and reverse primers, 2 μl of QIAGEN enzyme mix, 2 μl of QIAGEN dNTP mix, 17.5 μl nuclease-free water, and a 10 μl RNA input. The final concentrations for target primers and RNase P primers were set at 300 nM and 50 nM, respectively. Thermal cycling conditions comprised reverse transcription at 50 °C for 30 min, initial PCR activation at 95 °C for 15 min, followed by 40 cycles at 94 °C for 30 s, 58 °C for 30 s, and 72 °C for 30 s. Reactions were stopped by dropping the temperature to 4 °C. Please refer to Supplementary Table 1 and the Supplementary Data 1 spreadsheet for details on the primer sequences used in each BBP panel.

### Standard BioTools BiomarkX detection.
Fluidigm detection was performed as described previously with major updates on data collection and processing with the newer Fluidigm model, the BiomarkX[35]. The Cas13 detection reactions were divided into two distinct mixes—namely, the assay mix and the sample mix (details can be found below). These mixes were loaded onto a 192.24 microfluidic

Integrated Fluidic Circuit (IFC) for analysis with BiomarkX or Bio-markHD (Fluidigm instruments).

Assay mix: The assay mix comprised LwaCas13a (GenScript) at 20 nM final concentration. The assay also included 2X Assay Loading Reagent (Fluidigm), 69 U T7 RNA Polymerase mix (NEB), and crRNAs at 1 μM concentrations, resulting in a total volume of 16 μl per reaction.

Sample mix: The sample mix contained 1x homemade 10X clea-vage buffer (1 M Tris-HCl (pH 7.5), 0.1 M dithiothreitol, and nuclease-free water), 25.2 U RNase Inhibitor (New England Biolabs), 0.1X ROX reference dye (Invitrogen), 20X loading reagent (Fluidigm), 1 mM rNTPs (New England Biolabs), 9 mM MgCl$_2$ in water, and a 500 nM quenched synthetic fluorescent RNA reporter (FAM/rUrUrUrUrUrUrU/3IABkFQ/; Integrated DNA Technologies) was used for a total volume of 14 μl per reaction.

IFC loading and running on Fluidigm instruments: 192.24 IFCs were prepared and loaded with samples as described previously[35]. The IFCs were run on the BiomarkHD and the BiomarkX (all subsequent data) according to the manufacturer's instructions. For the Bio-markHD, the IFC was loaded onto the IFC Controller RX (Fluidigm) where the Load Mix script was run. For the BiomarkX, the IFC was loaded directly into the instrument. After proper IFC loading, images were collected over a 1-h period at 37 °C using a custom protocol pre-loaded into the Fluidigm's instruments.

Fluidigm data analysis: Fluidigm data analysis was performed as described previously[35]. Briefly, fluorescence analysis involved plotting reference-normalized and background-subtracted values for guide-target pairs. The reference-normalized value for a guide-target pair (at time point t and target concentration) was calculated as $(median(P_t - P_0) / (R_t - R_0))$, where $P_t$ is the guide signal (FAM) at the time point, $P_0$ is its background measurement, $R_t$ is the reference signal (ROX) at time point $t$, $R_0$ is its background measurement, and the median is taken across replicates. The same calculation was applied to the no-template control (NTC) of the guide, providing a background fluorescence value for the guide at time point $t$. The reference-normalized, background-subtracted fluorescence for a guide-target pair is the difference between these two values. A sample was considered positive if the signal produced was greater than the average signal produced by that assay's NTC plus three times the standard deviation of the signal in the NTCs.

**Designs and development of bloodborne pathogen panels**
**Designs.** Oligonucleotide primers and crRNA guides were designed to detect conserved regions. Briefly, these conserved regions were from the following 23 pathogens: (1) *Crimean Congo Hemorrhagic fever Virus* (CCHFV), (2) *Zaire ebolavirus* (EBOV), (3) *Human Immunodeficiency Virus 2* (HIV2), (4) *Human Immunodeficiency Virus 1* (HIV1), (5) *Lassa virus* (LASV), (6) *Marburg virus*, (7) *West-Nile Virus* (WNV), (8) *Yellow Fever virus* (YFV), (9) *Chikungunya virus* (CHI), (10) *Dengue virus* (DENV), (11) *Hantaan orthohantavirus* (HTV), (12) *Measles morbillivirus* (MMV), (13) *Rabies lyssavirus* (RBV), (14) *Rift Valley Fever virus* (RVFV), (15) *Zika virus* (Zika), (16) *Nipah henipavirus* (NPV), (17) *mpox virus* (MKP, formerly *Monkeypox virus*), (18) *Hepatitis B virus* (HBV), (19) *Hepatitis C virus* (HCV), (20) *Herpes Simplex Virus Type 2* (HSV2), (21) *Chlamydia trachomatis* (Chla), (22) *Treponema pallidum subsp. palli-dum* (Syph), and (23) *Trichomonas vaginalis* (Trich). If available, we leveraged pre-designed assays as published on the Activity-informed Design with All-inclusive Patrolling of Targets (ADAPT; https://adapt.run/) program. Otherwise, complete genomes were collected from National Center for Biotechnology Information (NCBI), aligned, and fed into ADAPT for crRNA design with >90% coverage. Highly conserved regions were selected, and primers were manually designed using NCBI's Primer-BLAST for optimal amplification target regions with crRNA binding regions in the middle. For our positive controls, we used RNase P primers and crRNA sequences as described previously[35]. All sequences used in this study can be found in Supple-mentary Data 1.

**In vitro analysis and limit of detection testing.** For each organism, five assays were designed and tested for specificity. The synthetic DNA targets contained the consensus sequence that was position-matched to the location of the BBP virus of interest targets in the viral genome. Samples were serially diluted down to a concentration of $10^6$ to $10^1$ copies/μL and were prepared for the specificity experiments according to the methods described above in Section 1.b. In-vitro transcription (IVT) of DNA targets.

Synthetic RNA of each BBP target ranging from $10^8$ to 10 copies/μL were analyzed on the CARMEN platform (QIAGEN amplification and Standard BioTools BioMarkX Detection) as well as by RT-qPCR (Applied Biosystems™ Power SYBR™ Green RNA-to-CT™ 1-Step Kit and 500 nM final concentration of appropriate forward and reverse primer pairs) as described in Section 1d. Utilizing the reference-normalized value of each guide-target pair, the threshold for each assay was calculated as (average(NTC) + 3*(standard deviation(NTC))). Through this threshold calculation, each sample was categorized as either negative or positive for each tested assay; whereas if a sample is above an assay threshold it is considered positive and if it is below then it is considered negative. The synthetic RNA of each target ranging from $10^8$ to 10 copies/μL was then evaluated against each guide specific threshold demonstrating the lowest concentration of detection per guide.

**Testing of contrived samples.** Genomic RNA/DNA from BEI Resources and Boston University's National Emerging Infectious Diseases Laboratories (NEIDL) were quantified using the Power SYBR™ Green RNA-to-CT™ 1-Step Kit (Applied Biosystems™). Reactions, performed in triplicate with 1 μL RNA input in 10-μL reactions, utilized a singleplex primer mix at 500 nM (see Supplementary Data 1 for sequences). Thermal cycling conditions included reverse transcription at 48 °C for 30 min, enzyme activation at 95 °C for 10 min, and 40 cycles at 95 °C for 15 sec and 58 °C for 1 min, followed by a melt curve step at 95 °C for 15 sec, 58 °C for sec, and 95 °C for sec. Standard curves were generated using spike-in RNA templates (generated as described in Section 1b) in tenfold serial dilutions and analyzed on the QuantStudio 6 Flex Real-Time PCR System. Contrived samples were created as previously described[35]. Briefly, 160 μL of pooled commercially available healthy normal blood serum samples (Boca Biolistics) were mixed with 40 μL of available nucleic acid extracts (BEI Resources and Boston Uni-versity's National Emerging Infectious Diseases Laboratories (NEIDL)) at known concentrations for a 200 μL total sample volume. Specifi-cally, the following reagent was obtained through BEI Resources, NIAID, NIH: RNA from Lassa Virus, Josiah, NR-31821. Following doc-umentation from the Zymo Research Quick-DNA/RNA™ MagBead kit, 200 μL of each contrived sample was utilized in the automated extraction on the ThermoFisher KingFisher Sample Purification Sys-tem (details of extraction process found above). Following automated extraction, nucleic acid extracts were tested in parallel; both on the CARMEN platform and validated commercially available detection kits (Aldatu Biosciences PANDAA LASV RT-qPCR for Lassa virus and Ray-Biotech Mpox virus (MPXV) PCR Nucleic Acid Detection Kit for mpox).Following specifications outlined from the manufacturers of the commercially available detection kits, results were determined and compared to the CARMEN output.

Contrived samples for LOD testing for LASV and MPOX were created as previously described[35]. Briefly, 160 μL of pooled commer-cially available healthy normal blood serum samples (Boca Biolistics) were mixed with 40 μL of available nucleic acid extracts (BEI Resour-ces) for a 200 μL total sample volume at various concentrations. Specifically, the following reagent was obtained through BEI Resour-ces, NIAID, NIH: Genomic DNA from Monkeypox Virus, hMPXV/USA/MA001/2022 (Lineage B.1, Clade IIb), NR58710, and the following reagent was obtained through BEI Resources, NIAID, NIH, as part of the WRCEVA program: Genomic RNA from Lassa Virus, Josiah, NR-50804.

Following documentation from the Zymo Research Quick-DNA/RNA™ MagBead kit, 200 μL of each contrived sample was utilized in the automated extraction on the ThermoFisher KingFisher Sample Purification System (details of extraction process found above). Following automated extraction, nucleic acid extracts were tested in parallel; both on the CARMEN platform and validated commercially available detection kits (Aldatu Biosciences PANDAA LASV RT-qPCR for Lassa virus and RayBiotech Mpox virus (MPXV) PCR Nucleic Acid Detection Kit for mpox). Following specifications outlined from the manufacturers of the commercially available detection kits, results were determined and compared to the CARMEN output.

Contrived samples for proof of concept (POC) testing were created as previously described[35]. Briefly, 160 μL of pooled commercially available healthy normal blood serum samples (Boca Biolistics) was mixed with 40 μL of available nucleic acid extracts (BEI Resources) for a 200 μL total sample volume. Specifically, the following reagents were obtained through BEI Resources, NIAID, NIH: RNA from Zaire Ebolavirus, Mayinga, NR 31806, RNA from Crimean-Congo Hemorrhagic Fever Virus, IbAr10200, NR-37382, RNA from Marburg Marburgvirus, Voege (German Voege), NR-31815, Genomic RNA from Chikungunya Virus, 181/25, NR-50345, Genomic RNA from Dengue Virus Type 1, Hawaii, NR-4287, Genomic RNA from Dengue Virus Type 3 (DEN-3), Philippines/H87/1956, NR-2771, Genomic RNA from Dengue Virus Type 4, H241 (Tissue Culture Adapted), NR-4289, RNA from Rift Valley Fever Virus, ZH501, NR-37379, Genomic DNA from Monkeypox Virus, hMPXV/USA/MA001/2022 (Lineage B.1, Clade IIb), NR58710. Additionally, the following reagent was obtained through BEI Resources, NIAID, NIH, as part of the WRCEVA program: Genomic RNA from Lassa Virus, Josiah, NR-50804. In the event that a gRNA/DNA sample could not be obtained, 160 μL of pooled commercially available healthy normal blood serum samples (Boca Biolistics) was mixed with 40 μL of synthetic pathogen material at known concentrations for a 200 μL total sample volume with a concentration of $10^5$ copies/μL. The following blood serum samples were obtained from Boca Biolistics for testing: Respiratory Syncytial Virus Blood Serum Sample, Severe acute respiratory syndrome coronavirus 2 Serum Sample, and Healthy Normal Blood Serum Sample. Following documentation from the Zymo Research Quick-DNA/RNA™ MagBead kit, 200 μL of each contrived sample was utilized in the automated extraction on the ThermoFisher KingFisher Sample Purification System (details of extraction process found above). Following automated extraction, nucleic acid extracts were tested on the CARMEN platform.

**Patient specimen validation.** Aiming to demonstrate the specificity and sensitivity of the CARMEN platform on confirmed-positive patient samples, clinical Gonorrhea, Lassa, and MPOX samples were tested. Gonorrhea samples were provided by the Massachusetts General Hospital. Lassa- and MPOX-positive samples were provided by the Institute of Genomics and Global Health (IGH), located in Ede, Osun State, Nigeria. All samples were tested on the CARMEN platform and, with the exception of Gonorrhea, samples were validated with commercially available detection kits (RealStar Lassa virus RT-qPCR Kit 2.0 for Lassa virus and RayBiotech Mpox Virus (MPXV) PCR Nucleic Acid Detection Kit for Monkeypox). Utilizing threshold parameters detailed in Section 1.e. In vitro analysis, results for each sample were determined on the CARMEN platform. Dually, using specifications from the manufacturers of the RT-qPCR detection kits, results were determined and compared to the CARMEN output.

**Ethics statement**
All research complied with relevant ethical regulations and institutional policies. *Neisseria gonorrhoeae* specimens were obtained from participants enrolled in a Massachusetts General Brigham (MGB) clinical trial (NCT05564299) and used under MGB IRB protocols 2019P003305 and 2020P000323. Work involving lassa virus and mpox specimens from the Institute for Genomic Health (IGH) was conducted

at IGH under approval from the National Health Research Ethics Committee of Nigeria (NHREC/01/01/2007) and at the Broad Institute under approval from the Harvard Longwood Campus Institutional Review Board (IRB24-0563). Additional local oversight was provided by the Irrua Specialist Teaching Hospital (ISTH) IRB (ISTH/REC/20221811/434). For primary collections, written informed consent was obtained from all participants. For secondary use of previously collected samples/data, a waiver of consent was reviewed and approved by the appropriate institutional review board/ethics committee.

**Reporting summary**
Further information on research design is available in the Nature Portfolio Reporting Summary linked to this article.

## Data availability
Source data are provided with this paper.

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

## Acknowledgements

We thank J.A.S., A.Si., and H.M. for helping with guide design, sequence analysis, or sharing reagents. This work is made possible by support from Flu Lab and a cohort of generous donors through TED's Audacious Project, including the ELMA Foundation, MacKenzie Scott, the Skoll Foundation, and Open Philanthropy. P.C.S. was supported by the Howard Hughes Medical Institute and Merck KGaA Future Insight Prize.

## Author contributions

M.K. and P.C.S. initially conceived this study and then involved L.K., K.M. and E.S. M.K., E.S., and L.K. designed the primers and crRNAs for the pathogens presented in this study. N.W. designed and performed the CARMEN RVP experiments and optimization on the BiomarkX. M.K., L.K., and K.M. performed the initial BBP experiments on the BiomarkX. K.M. conducted the Lassa and monkeypox contrived sample testing in an academic setting. G.S. and L.A-B. performed the clinical evaluation of the CARMEN BBPs at MGH under guidance from M.K., K.M. and J. L. K.M. and L.S. conducted the clinical evaluation of the CARMEN BBPs at IGH in Nigeria with assistance from P.E., A.M.I., A.E.S., O.O.O., A.O.A., C.l'A., I.B., M.F.P., C.W. and supervision from M.K., E.S., A.O., C.H., and P.C.S. M.K. generated the figures with help from K.M. and wrote the paper with help from K.M. and E.S., and guidance from P.C.S. All authors reviewed the manuscript.

## Competing interests
