## [Transparent Peer Review file · Nature Communications]

Multiplexed detection of febrile infections using CARMEN

Corresponding Author: Professor Pardis Sabeti

Version 0:

Reviewer comments:

Reviewer #1

(Remarks to the Author)

The authors have addressed most of my comments, and the manuscript has significantly improved and fits Nature Comms. Therefore, I recommend publication. I just recommend double-checking the number of samples in figure 4. For example, for LASV, the concordance table indicates a higher number of samples than are shown in the corresponding heatmap. Make sure to add data from the missing samples in this figure or reference the supplementary figure, where the remaining samples can be found.

Reviewer #2

(Remarks to the Author)

In the revised manuscript, the authors have provided additional supporting data and addressed the previous concerns. The manuscript is now suitable for publication in Nature Communications.

Reviewer #3

(Remarks to the Author)

Dear Dr. Kamariza, Dr. Sabeti, and colleagues,

Thank you for your thoughtful and thorough responses to the reviewers' comments. I appreciate the care and scientific rigor you showed in addressing each point raised during the first round of review.

Your manuscript represents a meaningful advancement in the field of CRISPR-based diagnostics by extending the CARMEN platform's capabilities into new domains of clinical relevance. The development and validation of multiplexed bloodborne pathogen panels (BBP1–3), covering viral hemorrhagic fevers, mosquito-borne viruses, and sexually transmitted infections, is both timely and commendable.

Several aspects of your revised submission particularly stand out:

1. Expanded validation and analytical depth: The additional LOD experiments with 20 replicates per target and comparisons to RT-qPCR kits provided compelling evidence of CARMEN's superior sensitivity and reproducibility, especially at low copy numbers.
2. Specificity and assay refinement: The careful removal of the *Trichomonas* assay based on off-target detection and the robust cross-reactivity assessments demonstrated a high standard of assay curation and validation.
3. Field deployment and practical impact: Validation of the platform using confirmed-positive patient samples from both the U.S. and Nigeria, including LASV and MPOX cases, highlights the global relevance and deployability of your system. Limitations were discussed, and a structured roadmap was proposed to guide future enhancements of the system.
4. Clarification of methodological details: Your responses clarified critical elements such as the distinction between RT-PCR and RT-qPCR, data normalization, cost analysis, and the rationale for discordant sample interpretation.

5. Contextualization within the literature: The expanded discussion more clearly differentiates this work from your prior publications and places it in an appropriate context alongside other CRISPR-based platforms such as SHERLOCK, SHINE, and DETECTR.

The manuscript is now clear, comprehensive, and scientifically robust. It also makes a significant contribution to the translational progress of CRISPR diagnostics, providing a scalable and sensitive platform for high-throughput pathogen detection across various clinical environments.

I have no further comments and fully support the publication of this work.

Reviewer #4

(Remarks to the Author)

Reviewer #5

(Remarks to the Author)

Reviewers' Comments

Reviewer #1:

The authors have addressed most of my comments, and the manuscript has significantly improved and fits Nature Comms. Therefore, I recommend publication. I just recommend double-checking the number of samples in figure 4. For example, for LASV, the concordance table indicates a higher number of samples than are shown in the corresponding heatmap. Make sure to add data from the missing samples in this figure or reference the supplementary figure, where the remaining samples can be found.

We thank the reviewer for their thoughtful and constructive comments which help us improve our manuscript. Regarding **Figure 4**, we mistakenly included 54 negative samples on the concordance sheet for LASV CARMEN v qPCR results. We have since corrected this number to 59, which now totals 90 samples tested across both assays. We have updated **Figure 4F** in the manuscript (also shown below).

Figure 4. Sensitive, accurate, detection of Gonorrhea, Chlamydia, Lassa virus, and Monkeypox in patient samples. (A-B) Heatmaps illustrating CARMEN detection of (A) *Neisseria gonorrhoeae*, and (B) *Chlamydia trachomatis*. (C) Comparison analysis of CARMEN fluorescence to RT-qPCR Ct values for Gon patient samples. (D-E) Heatmaps showing fluorescence values of (D) LASV or (E) MPOX clinically positive patient samples. For all heatmaps, normalized fluorescence signal at 1 h post-reaction initiation of corresponding viral Cas13 crRNAs is shown. CPC = Combined Positive Control. NDC = No Detection Control. (F-G) Concordance table of CARMEN fluorescence to RT-qPCR Ct values for (F) LASV and (G) MPOX patient sample. (H-I) Comparison analysis of CARMEN fluorescence to RT-qPCR Ct values for (H) LASV, and (I) MPOX.

Reviewer #2:

In the revised manuscript, the authors have provided additional supporting data and addressed the previous concerns. The manuscript is now suitable for publication in Nature Communications.

We thank the reviewer for their thoughtful and constructive comments which helped us improve our manuscript. We are especially grateful for their recognition of the study's significance and its implications for public health surveillance and clinical diagnostics. Thank you!

Reviewer #3:

Dear Dr. Kamariza, Dr. Sabeti, and colleagues,

Thank you for your thoughtful and thorough responses to the reviewers' comments. I appreciate the care and scientific rigor you showed in addressing each point raised during the first round of review.

Your manuscript represents a meaningful advancement in the field of CRISPR-based diagnostics by extending the CARMEN platform's capabilities into new domains of clinical relevance. The development and validation of multiplexed bloodborne pathogen panels (BBP1–3), covering viral hemorrhagic fevers, mosquito-borne viruses, and sexually transmitted infections, is both timely and commendable.

Several aspects of your revised submission particularly stand out:

1. Expanded validation and analytical depth: The additional LOD experiments with 20 replicates per target and comparisons to RT-qPCR kits provided compelling evidence of CARMEN's superior sensitivity and reproducibility, especially at low copy numbers.
2. Specificity and assay refinement: The careful removal of the *Trichomonas* assay based on off-target detection and the robust cross-reactivity assessments demonstrated a high standard of assay curation and validation.
3. Field deployment and practical impact: Validation of the platform using confirmed-positive patient samples from both the U.S. and Nigeria, including LASV and MPOX cases, highlights the global relevance and deployability of your system. Limitations were discussed, and a structured roadmap was proposed to guide future enhancements of the system.
4. Clarification of methodological details: Your responses clarified critical elements such as the distinction between RT-PCR and RT-qPCR, data normalization, cost analysis, and the rationale for discordant sample interpretation.
5. Contextualization within the literature: The expanded discussion more clearly differentiates this work from your prior publications and places it in an appropriate context alongside other CRISPR-based platforms such as SHERLOCK, SHINE, and DETECTR.

The manuscript is now clear, comprehensive, and scientifically robust. It also makes a significant contribution to the translational progress of CRISPR diagnostics, providing a scalable and sensitive platform for high-throughput pathogen detection across various clinical environments.

I have no further comments and fully support the publication of this work.

We sincerely thank the reviewer for their thoughtful and generous evaluation, which helped improve our manuscript. We appreciate the reviewer's acknowledgment of the engineering rigor behind our workflow, spanning nucleic acid extraction, amplification, and detection, and we are encouraged by their assessment of the manuscript as a meaningful addition to the field of fluorescence-based multiplex diagnostics. We are especially grateful for the recognition of the technical innovations presented in this study, as well as the reviewer's enthusiasm for the potential diagnostic applications of the CARMEN platform. Thank you!